# Competition between proton transfer and intermolecular Coulombic decay in water

Clemens Richter [1], Daniel Hollas[2], Clara-Magdalena Saak [3], Marko Förstel[4,7], Tsveta Miteva[5], Melanie Mucke [3], Olle Björneholm[3], Nicolas Sisourat[5], Petr Slavíček[2] & Uwe Hergenhahn [1,6]

Intermolecular Coulombic decay (ICD) is a ubiquitous relaxation channel of electronically excited states in weakly bound systems, ranging from dimers to liquids. As it is driven by electron correlation, it was assumed that it will dominate over more established energy loss mechanisms, for example fluorescence. Here, we use electron–electron coincidence spectroscopy to determine the efficiency of the ICD process after $2a_1$ ionization in water clusters. We show that this efficiency is surprisingly low for small water clusters and that it gradually increases to 40–50% for clusters with hundreds of water units. *Ab initio* molecular dynamics simulations reveal that proton transfer between neighboring water molecules proceeds on the same timescale as ICD and leads to a configuration in which the ICD channel is closed. This conclusion is further supported by experimental results from deuterated water. Combining experiment and theory, we infer an intrinsic ICD lifetime of 12–52 fs for small water clusters.

[1] Leibniz Institute of Surface Engineering (IOM), Permoserstr. 15, 04318 Leipzig, Germany. [2] Department of Physical Chemistry, University of Chemistry and Technology Prague, Technická 5, 16628 Prague 6, Czech Republic. [3] Department of Physics and Astronomy, Uppsala University, Box 516751 20 Uppsala, Sweden. [4] Max Planck Institute for Plasma Physics, Boltzmannstr. 2, 85748 Garching, Germany. [5] Laboratoire de Chimie Physique Matière et Rayonnement, UMR 7614, Sorbonne Université, CNRS, F-75005 Paris, France. [6] Max Planck Institute for Plasma Physics, Wendelsteinstr. 1, 17491 Greifswald, Germany. [7] Present address: Institute for Optics and Atomic Physics, Technical University Berlin, Hardenbergstr. 36, 10623 Berlin, Germany. These authors contributed equally: Clemens Richter, Daniel Hollas.  Correspondence and requests for materials should be addressed to P.Síče. (email: petr.slavicek@vscht.cz) or to U.H. (email: uwe.hergenhahn@ipp.mpg.de)

The pioneering work on intermolecular Coulombic decay (ICD) by Cederbaum and co-workers[1] has initiated continuing interest in non-local autoionization processes in weakly bound atomic and molecular systems. In ICD, a positively charged electron hole, created in an inner valence or core electronic level, is refilled by a valence electron, and the excess energy is released by ejecting a second electron from a neighboring unit. The final state of the process is thus represented by two separate singly charged atomic or molecular sites instead of one doubly charged site, found in a regular Auger process. Depending on the constituents of the system under consideration, ICD was taken to mean either Intermolecular or Interatomic Coulombic Decay.

ICD was first observed in rare gas clusters following inner-valence electron ionization[2–4]. Later, ICD in the water dimer and water clusters was unequivocally identified by coincidence spectroscopy, again following inner-valence ionization[5,6]. Related radiationless decay processes were also found in the liquid phase after core ionization[7–9]. For reviews of the field see refs.[10,11]. Non-local electronic decay channels are an important, yet so far largely unconsidered component in photochemistry and radiation chemistry[8,12]. The potential impact of ICD in the context of oxidative stress and radiation treatment has been discussed[13–15] because highly damaging slow electrons and two radical cations are formed by the ICD process.

The yield of ICD is determined by its decay rate relative to other relaxation mechanisms, such as radiative transitions or internal conversion (IC). For rare gas clusters, the ICD lifetime (inverse rate) was found to be on the order of tens to hundreds of femtoseconds upon inner-valence ionization[16,17]. On this timescale, radiative transitions do not play an important role, unless the decay of core holes in heavier elements is considered. Consequently, ICD was experimentally found to quench all other relaxation channels of rare gas clusters[18,19]. On the other hand, significant nuclear motion may occur in molecular systems even on the femtosecond timescale[20,21]. Therefore, the ICD process cannot be described without considering nuclear dynamics. A coupling between nuclear and electronic motion was observed, for example, for the Ne–He dimer[22] and for aqueous systems upon core ionization[8,9,23–26]. Depending on the particular situation, the nuclear dynamics may suppress or accelerate the interatomic or intermolecular autoionization processes.

In the present work, we investigate the coupling of electron and nuclear dynamics in water clusters in detail. While water clusters provide a suitable experimental environment that allows for a detection of electrons and/or molecular fragments, they also provide a link from very small systems to bulk water[27,28]. Here, we consider $2a_1$ photoionization of water clusters of different sizes, followed by ICD. The process can be written as follows:

$$H_2O \cdots H_2O \xrightarrow[-e1]{h\nu} H_2O \cdots H_2O^+ \left(2a_1^{-1}\right)$$
$$\xrightarrow[-e2]{} H_2O^+ \cdots H_2O^+ \rightarrow H_2O^+ + H_2O^+ \quad (1)$$

We also need to consider the nuclear dynamics. Both valence[29] and core[25] ionization initiate a proton transfer along the hydrogen bond coordinate:

$$H_2O \cdots H_2O \xrightarrow[-e1]{h\nu} H_2O \cdots H_2O^+ \rightarrow H_3O^+ \cdots OH \quad (2)$$

Immediately, the question arises whether these two processes interfere. Does the proton transfer also take place upon $2a_1$ ionization? If so, does it lead to an enhancement of the ICD rate as seen upon core ionization[9,25] or does it close the ICD channel instead?

In fact, there are indications that the efficiency of the ICD process is lower than unity. Comparing the simulated abundance of $H_2O^+$ fragments after ionization of water dimers with an electron impact experiment[30], Svoboda et al.[29] argued that some production of water cations via ICD after inner-valence ionization takes place, but likely only for a fraction of these ionized states. Examples of proton transfer faster than the measured ICD rate for a neon dimer[16] (isoelectronic with water) were found[8,29]. Therefore, it does not seem probable that ICD fully dominates inner-valence relaxation for the case of water dimers.

In the present work, we address this issue by a combination of simulation and experimental tools. We experimentally determine the efficiency of the ICD process in water cluster ensembles by quantitative analysis of the photoelectron-ICD electron coincidence signal[18]. Furthermore, we performed the experiments for regular water clusters as well as for their deuterated analogs. The isotopic substitution reliably reveals the role of the nuclear dynamics in the electronic decay processes[25]. We explicitly calculate the proton transfer process initiated by the $2a_1$ electron ionization. The efficiency data together with theory can be translated into an estimate for the ICD lifetimes. These estimates are compared with lifetimes calculated with the Fano-CI method[31] (see Methods section).

Additionally, our electron–electron coincidence data for water clusters show the ICD efficiency as a function of cluster size. The dependence of the ICD rate on the number of neighboring atoms or molecules is an important problem that so far has only been studied for rare gas clusters. Theoretically, small clusters of up to one coordination shell were considered[32,33], while experimentally it could only be shown that the rate for Ne $2s$ ICD of bulk sites exceeds the one of surface states[4]. In this work, we have identified several size-dependent factors that have an influence on the ICD rate, and will discuss them in detail. Common to all system sizes we find that proton transfer away from the $2a_1^{-1}$ ionized site occurs on a timescale comparable to ICD, and leads the system into a region of coordinate space in which the ICD channel is no longer open.

## Results

**Experimental results.** In our experiment, we irradiated cluster ensembles with mean sizes $\langle N \rangle$ ranging from 5 to 246 water molecules at photon energies sufficient to induce the ICD process, that is larger than the binding energy $E_b$ of the $2a_1$ level (32.0 eV in $\langle N \rangle = 100$ clusters[27]). Cluster sizes were determined from the operation parameters of the cluster source, compiled in Supplementary Table 1, by a procedure detailed in Supplementary Methods. Coincident detection enabled the observation of electron pairs, consisting of a photoelectron together with the pertaining ICD electron. An example for a two-dimensional histogram giving their kinetic energies is shown as Supplementary Fig. 1. From these data, we extracted an experimental measure of the ICD efficiency $\alpha_{ICD}$, defined as the fraction of $2a_1$-ionized states which decay by ICD. Our analysis follows previous work on coincident spectra from Ne clusters, in which $\alpha_{ICD}$ after $2s$ photoionization is 100%[18]. We assume that $\alpha_{ICD}$ under ideal conditions is described by the ratio of coincident electron pairs $P$ $(E_{ph}, E_{ICD})$ to the total number of photoelectrons detected upon photoionization of the $2a_1$ state, $p(E_{ph})$, where $E$ denotes the kinetic energy of the respective electron, and $P$, $p$ the coincident and non-coincident event rates for electrons in the given intervals of kinetic energy. To get quantitatively correct results for actual experimental conditions (see the Methods section), some more parameters have to be included:

$$\alpha_{ICD} = \frac{P\left(E_{ph}, E_{ICD}\right)}{p\left(E_{ph}\right)} \frac{1}{c\,\gamma(E_{ICD})}. \quad (3)$$

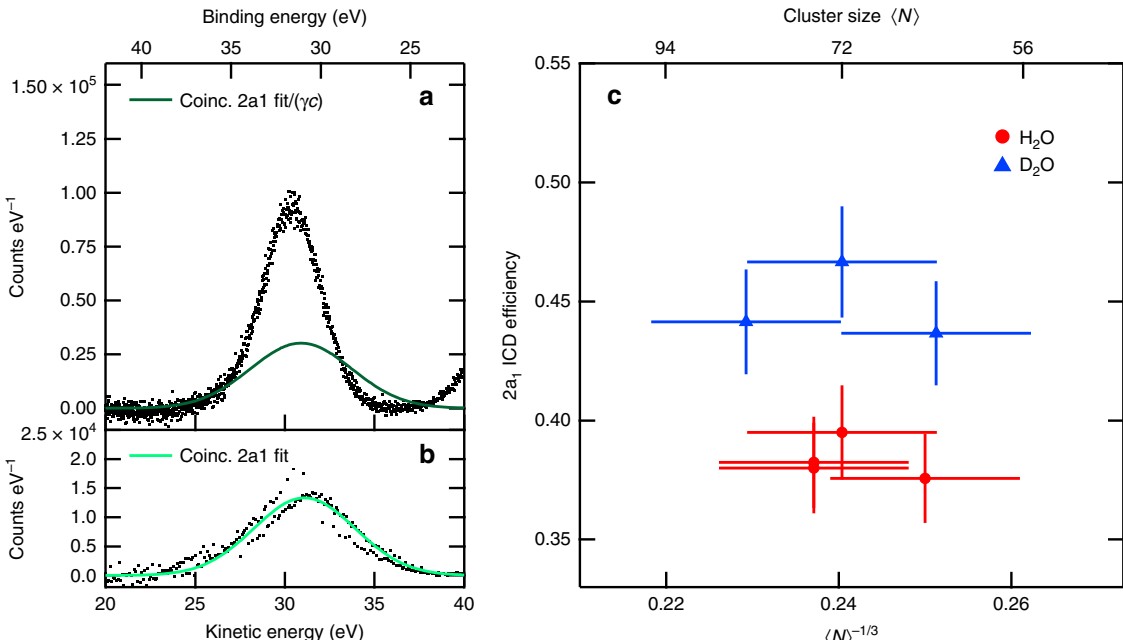

**Fig. 1** Efficiency of ICD after $2a_1$ ionization of water clusters. **a** Photon excited electron spectrum of a water cluster jet in the region of $2a_1$ binding energies. The mean cluster size is $\langle N \rangle = 76$, the photon energy $h\nu = 62$ eV. A linear background was subtracted. Electron energy is represented as kinetic energy KE (bottom axis) or binding energy BE (top axis). **b** Kinetic energy spectrum of photoelectrons detected in coincidence with an ICD electron, after background subtraction. **a**, **b** are derived from the same data set. The solid trace in **b** is a fit to the coincident data. For comparison with the undiscrimated data, the curve is multiplied by the inverse of the detection efficiency $\gamma(E_{ICD})$ and the degree of condensation $c$, with values of 0.58(4) and 0.76(5) in this example; the result is shown in **a** (solid trace). **c** Effect of isotopic substitution on the ICD efficiency, measured at $h\nu = 60$ eV. Error bars shown here represent the standard deviation due to stochastic errors. Systematic errors ('scale errors') in $\langle N \rangle$ and $\alpha_{ICD}$ (see below and Supplementary Notes 1 and 2) exist also but influence values for $H_2O$ and $D_2O$ equally

$c$ is the degree of condensation in the water cluster jet, and $\gamma(E_{ICD})$ the detection efficiency for secondary electrons. Here, we neglect further potentially relevant factors such as inelastic scattering losses. We give a detailed discussion of these effects and a derivation of Eq. (3) in Supplementary Methods. $c$ was determined at each data point from outer valence electron spectra, with an example shown in Supplementary Fig. 2, and corresponding results in Supplementary Fig. 3.

A background correction is essential when determining $P(E_{ph}, E_{ICD})$ and $p(E_{ph})$. This background is comprised of inelastically scattered electrons as well as electron intensity from direct and indirect double ionization of the gaseous water molecules in the cluster jet[34,35], and overlaps with the $2a_1$ peak and the ICD signature. After assessing all parameters, we arrived at the results summarized in Fig. 1.

In Fig. 1, the data in panels (a) and (b) correspond to $p(E_{ph})$ and to $P(E_{ph}, E_{ICD})$, integrated over $E_{ICD}$ in the interval 0–5 eV, respectively. For an ideal detector, fully condensed cluster jet and ICD efficiency of one, the two panels would be identical. In the actual experimental data, any mismatch between panels (a) and (b) after correcting for finite detection efficiency and finite degree of condensation can be interpreted as $2a_1$ photoionization events that do not result in the emission of an ICD electron. From the comparison of the solid curve in Fig. 1a with the non-coincident data, we can therefore directly infer that the ICD efficiency is lower than unity.

Figure 1c shows values for the ICD efficiency determined from quantitatively interpreting the ratio of coincident to total $2a_1$ ionization events (see Supplementary Methods for details). Here, we include results for clusters of water and of isotopically substituted water ($D_2O$) obtained with otherwise identical settings

of the cluster source. While the experimental results are subject to non-negligible uncertainties, the following two main points are clearly observed: Firstly, the efficiency of the ICD process is well below unity and above that, there is a visible isotope effect, with $D_2O$ clusters exhibiting higher ICD efficiencies. We can thus safely conclude that the electronic ICD process in water clusters competes with other dynamical processes.

Figure 2 displays the resulting efficiency for all measured cluster sizes. We applied different routines for background correction, a simple linear model similar to the analysis in ref. [18] and a more sophisticated fit of Gaussian distributions on a linear baseline. Examples are shown in Supplementary Fig. 4. Empty and filled symbols show the efficiency derived by using the linear and Gaussian background correction, respectively. Both methods are reasonable approaches to peak-to-background separation in our spectra, and they (approximately) represent the lower and upper limits for values of the ICD efficiency compatible with our data. The graph shows an ICD efficiency well below unity for all cluster sizes measured, with values between 0.05 and 0.2 for the smallest sizes. In the following, we will first discuss the reason why the ICD efficiency generally is lower than one. The size dependence of the ICD efficiency will be addressed in the Discussion section.

Considering the clear isotope dependence and the fact that ICD is the only accessible electronic relaxation channel upon inner-valence ionization, proton transfer dynamics is the primary suspect for a competing relaxation process. The proton transfer reaction after valence ionization takes several tens of femtoseconds in liquid water[36] and molecular clusters[29] while it is much faster upon core ionization[8]. In the latter case proton transfer is known to compete with Auger decay and ICD, and in that

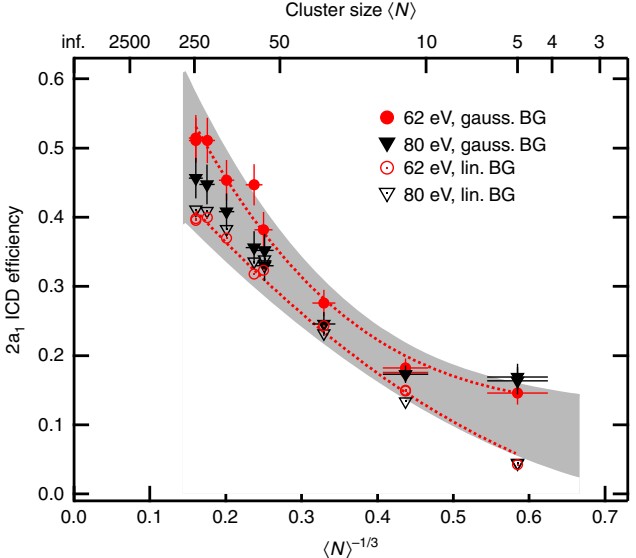

**Fig. 2** Dependence of the ICD efficiency on mean cluster size. The efficiency of the decay of $2a_1$ vacancies by ICD is shown as a function of mean cluster size $\langle N \rangle$ (top axis) or $\langle N \rangle^{(-1/3)}$ (bottom axis, with $\langle N \rangle^{(-1/3)} \propto$ to inverse cluster radius). Empty and filled symbols represent different methods of background subtraction. Error bars shown for some of the data represent the standard deviation due to stochastic errors. Dotted lines show polynomial fits to the 62 eV data sets, to guide the eye. The effect of systematic error on these two curves is shown by the shaded region

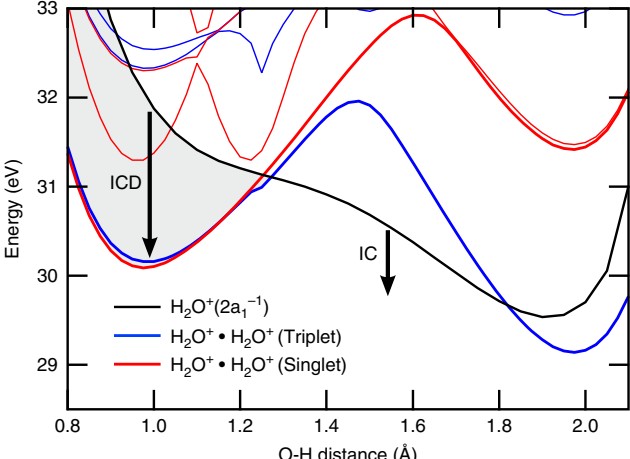

**Fig. 3** Calculated potential energy curves. The potential energy curves of the singly (black) and doubly ionized singlet (red) and triplet (blue) states for water dimer are shown along the proton transfer coordinate. The shaded area denotes the region in which ICD is energetically allowed and arrows indicate relaxation via ICD or internal conversion (IC), respectively. The $2a_1$-ionized state was calculated at the MS-CASPT2(15, 8)-SA8/6-31++g** level, the triplet doubly ionized states at the MS-CASPT2(6, 4)-SA6/6-31++g** level and the singlet doubly ionized states at the MS-CASPT2(6, 4)-SA10/6-31++g** level. Absolute energies were calculated relative to the minimum on the electronic ground state of the water dimer. The reference $2a_1$ ionization energy and the lowest triplet electronic state of the dication were calculated at the MP4/aug-cc-pVTZ level of theory. More details are given in the Methods section. A Mulliken population analysis shows that the positive hole is completely localized on the donor water unit when the $2a_1$-ionized state is created. After the proton transfer, the positive charge is mostly localized on the $H_3O$ moiety ($+0.83e$)

context it was termed proton transfer-mediated charge separation (PTM-CS)[8,9].

**Theoretical results.** Figure 3 shows that the proton transfer reaction is open upon inner-valence ionization. The figure displays the potential energy curve along the proton transfer coordinate (O–H distance) of a water dimer for the singly ionized $2a_1^{-1}$ state, which has an initial energy of ca. 32 eV above the neutral ground state. The negative slope of this state indicates that the proton transfer process is energetically allowed. In addition, the figure shows the few lowest singlet (red) and triplet (blue) states of the doubly charged water dimer. Both singlet and triplet doubly ionized ground states correspond to the final states of the ICD process, which are charge separated or '1h1h' states. The corresponding potential energy curves exhibit a double-well shape: The left minimum correspond to the $H_2O^+ \cdots H_2O^+$ structure while the right one represents the $OH^+ \cdots H_3O^+$ structure. The maximum on the curves correspond to the transient $HO \cdots H^+ \cdots H_2O^+$ species. From the energetical ordering of the curves, it is immediately clear that autoionization is energetically open in the ground state geometry of the water dimer ($r(OH) \sim 0.97$ Å). However, even a modest geometry change along the OH axis can close the autoionization channel. From our calculations we infer that the ICD channel closes for proton-transferred structures with a crossover distance of ~1.3 Å. Proton transfer thus represents a viable competing mechanism for the ICD process.

The energy difference between the singly ionized $2a_1^{-1}$ state and the doubly ionized final state gives the kinetic energy of the emitted ICD electron. Within our level of theory (as presented in Fig. 3), we deduce an ICD electron distribution of approximately 0 to 2 eV, which is in good agreement with previously reported experimental results for the water dimer[5]. The exact energetics and crossover distance depend on the level of theory (see Supplementary Note 2), but the mechanism is clear. The complete closing of the ICD channel upon proton transfer is also supported by the electron–ion–ion coincidence experiment of Jahnke et al.[5], in which no ICD-induced break-up into $OH^+ \cdots H_3O^+$ was detected.

The one-dimensional cuts through the potential energy surface often provide a biased view on the dynamics of a system. To find out more about the relaxation pathways following $2a_1$ ionization, we have performed ab initio molecular dynamics simulations of the water dimer. The dimer is exceptional among the water clusters, because only for this size the cluster contains two distinguishable entities: the proton donating and the proton accepting unit. In our calculations, we primarily focus on the dynamics following the inner-valence ionization of the proton donating unit, as essentially all water units act as hydrogen bond donors in larger clusters and in liquid water. Upon the ionization of the $2a_1$ electron on the proton donor, the proton transfer takes place for all trajectories and is essentially completed after ~8 fs. Figure 4 shows the decline of the ICD-capable population (that is the fraction of structures, for which the energy of the doubly ionized state is below the energy of the singly ionized state). According to the figure, the time available for ICD is limited to a few fs. In our further discussion, we approximate this temporal dependence by a step function with a characteristic time $t_{open}$, approximated as the time at which the ICD-capable population has halved. $t_{open}$ has a value of ~2.9 fs for regular and ~4.0 fs for deuterated water dimers. These calculations immediately elucidate the isotope effect seen in the experimental results. For deuterated water clusters, the ICD channel stays open over a longer time, which is sufficient to enhance the efficiency of the process significantly.

Combining the experimental information on the ICD efficiencies with the calculated results on the kinetics of the proton transfer reaction, we can estimate the ICD lifetime. We compare the smallest clusters measured, with an average of five water molecules, with the calculations for the water dimer. Further we assume that ICD can be described by an exponential law characterized by a single lifetime $\tau_{ICD}$. However, the ICD channel is energetically open only for a time $t_{open}$. Then, the efficiency of ICD is related to these quantities by:

$$\alpha_{ICD} = 1 - \exp\left(-t_{open}/\tau_{ICD}\right) \qquad (4)$$

The experimental efficiency $\alpha_{ICD}$ for the smallest $H_2O$ clusters measured is between 0.05 to 0.22, which corresponds to a $\tau_{ICD}$ of 57 to 12 fs within our model ($t_{open} = 2.9$ fs). This ICD lifetime

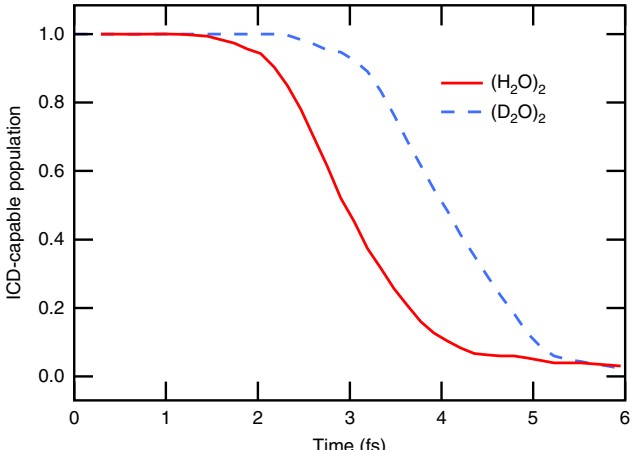

**Fig. 4** Closing of the ICD channel. Fraction of molecular dynamics trajectories in which the ICD channel is open, as a function of time after inner-valence ionization

should not depend on the isotopic substitution, as the process is of a purely electronic nature. Consequently, using the $\tau_{ICD}$ deduced from the $H_2O$ clusters together with $t_{open} = 4.0$ fs for deuterated water, we get an ICD efficiency which is ~1.35 times larger. This increase is in the same range, albeit somewhat larger, than the measured ICD enhancement for $\langle N \rangle = 60$–80 deuterated clusters, which is between 1.18 and 1.27, depending on the background model. We should however keep in mind that these estimates are simplified, as the ICD rate in fact depends on the geometry of the system.

Additionally, we have directly estimated the ICD rates using the Fano-CI theory at the equilibrium geometry of the water dimer (see Methods section). Assuming that only one singlet and one triplet final states are available, we obtained 72 and 131 fs for the lifetimes of the donor and acceptor water molecules, respectively, which is compatible with the experimental value of $150 \pm 50$ fs reported for the isoelectronic neon dimer[16]. An estimate for the lifetime in small water clusters can be extrapolated from these values by considering that each water molecule forms two donor and two acceptor bonds. Summing the respective decay rates, one obtains a total lifetime of 23 fs. Of course, this is again a simple estimate, but the value is in good agreement with the ICD lifetime extracted from the experiment for the smallest clusters (12–52 fs, see above). A calculation of the lifetimes as well as an estimate of the ICD efficiency as a function of the number of open ICD channels are presented in Supplementary Tables 2 and 3, and Supplementary Notes 2 and 3.

## Discussion

Our results have unambiguously shown that proton transfer strongly competes with the ICD process, limiting ICD to the first few femtoseconds upon inner-valence ionziation. As the inherent electronic ICD lifetime is much longer, this is reflected in a limited efficiency of the ICD process in finite size water aggregates. Earlier work by Jahnke et al.[5] clearly observed ICD of the water dimer, but was insensitive to the fraction of ionized states that decay via proton transfer, as this channel does not lead to break-up into two charged fragments. In a very recent study of

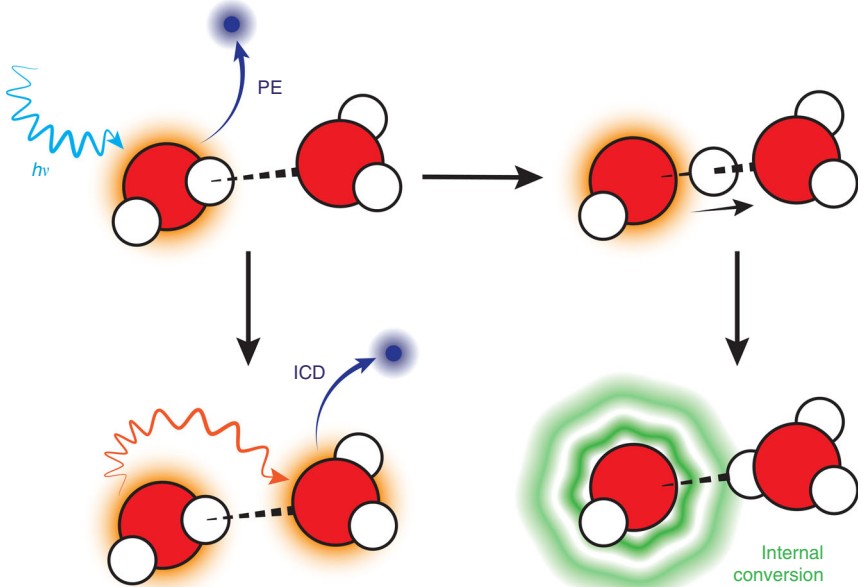

**Fig. 5** Processes and ionic fragments after $2a_1$-ionization of the water dimer. The system is photoionized at its ground state geometry into an electronically excited ionic state. Subsequently, it relaxes via ICD into a 1h·1h final state and at the same time evolves along the OH coordinate. The crossing of the singly excited state and the 1h·1h final state marks the point beyond which the ICD process is no longer viable. The proton transfer eventually leads to the formation of a highly excited OH radical and a hydronium cation $H_3O^+$

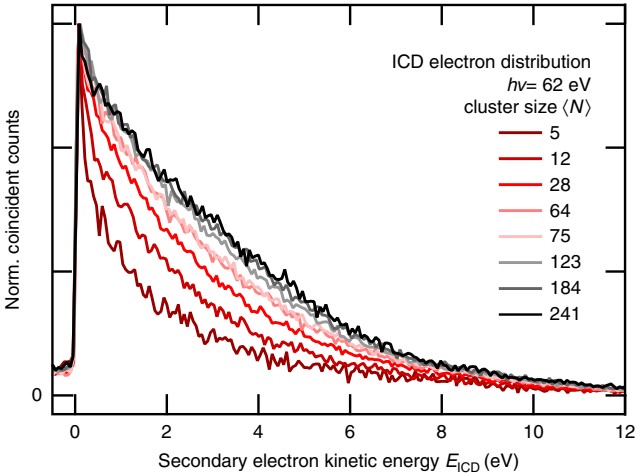

**Fig. 6** Energy distribution of ICD electrons. Experimental data for all mean cluster sizes measured, normalized to the same height and without background subtraction. Coincident counts were summed up between 26 and 35 eV photoelectron kinetic energy

water–tetrahydrofuran dimers, both ICD and proton transfer channels were observed, but no detailed discussion of the latter has been given[37].

The proton transfer process responsible for closing the ICD channel is analogous to proton transfer mediated charge separation (PTM-CS), recently identified in water upon core ionization[8,25]. However, the situations upon core and inner-valence ionization are dramatically different, because in the first case the autoionization is energetically allowed at each geometry. Proton transfer then enhances the probability of non-local autoionization processes[9]. In the second case, ICD is suppressed by the proton transfer, as we have shown in the present work.

It is also interesting to consider the consequences of the present work for radiation chemistry. Here, the creation of charged species on longer time scales, that is times at which both ICD and proton transfer have completed, is of importance. Figure 5 gives an illustrative overview of the processes following $2a_1$-ionization in the water dimer. While ICD creates two cations, after proton transfer the system remains singly charged, with a highly excited, neutral OH radical at the site of the original ionization, and the charge transferred to a neighboring molecule. The excited OH radical likely will release a hydrogen atom; after that the hydrogen atom can be transformed into a solvated electron and a proton, and the oxygen atom will react with water, forming either two OH radicals or hydrogen peroxide. The set of final products are thus similar to those formed upon ICD, yet, instead of a highly damaging slow electron[12], a hydrogen atom of relatively low potential for biological damage is formed. The results indicate that small strongly hydrogen-bonded systems are less susceptible to damage from slow autoionization after irradiation than previously thought.

Finally, we address the size dependence of the ICD efficiency shown in Fig. 2. The graph clearly shows an increase of the ICD efficiency from 0.05–0.2 to 0.4–0.5 with decreasing values of $\langle N \rangle^{-1/3}$ (increasing cluster size). This trend is a result of the combined effect of several factors:

With the increase in cluster size, electron polarization effects lead to a shift of the excited states towards lower binding energy[27]. This shift is larger for doubly ionized molecules. Consequently, the energy difference between singly and doubly ionized states increases[26], which, in turn, prolongs the time $t_{open}$ that the system is in a state open for the ICD process, and offers

more doubly ionized states to decay into. As a consequence, $\tau_{ICD}$ decreases. To better underpin these expectations, we have calculated the potential curves of singly inner-valence ionized and doubly ionized states shown in Fig. 3, using a water dimer embedded in a dielectric continuum as a simple approximation to a large water aggregate. Results are shown in Supplementary Fig. 5 and described in Supplementary Note 4. An increase of the energy difference between the states, compared to the case of small clusters, is apparent.

The energy shift is also clearly reflected in the experimental results. Figure 6 shows the normalized ICD electron distribution ($P(E_{ph}, E_{ICD})$, integrated over $E_{ph}$) for various cluster sizes as a function of kinetic energy. The distribution is very narrow for small clusters with five water units and resembles the energetics in Fig. 3 and ref. [5] With growing cluster size, we observe higher kinetic energies of the ICD electrons, indicating a bigger energetic gap between the $2a_1$-ionized and lowest doubly ionized states. From our polarizable continuum model, we estimated the bulk limit of this gap as ~6.5 eV, which is in reasonable agreement with the data in Fig. 6 given the crude nature of this model.

We expect this mechanism to be present, as we have both theoretical and experimental evidence for it. Nevertheless, two other factors might as well be important:

The number of final states accessible for ICD might not only increase due to energetical shifts, but also due to an increase in the average number of neighboring water units for water centers in a larger cluster. In calculations for rare gas clusters, a dramatic decrease of lifetime was predicted due to this effect[32,33]. In water, we expect a lesser effect because even in a large water network, every molecule is at most fourfold coordinated. An inspection of calculated water clusters structures[38] up to $(H_2O)_{21}$ shows a very gradual increase to an average of 3.2 bonds per molecule.

And finally, the hydrogen bond length decreases in larger clusters (the O–O distance is 2.9 Å in the water dimer and around 2.8 Å in bulk water[39]). In principle, this leads to both faster ICD because of larger orbital overlap and faster proton transfer. A Fano-CI calculation we did for the dimer with the O–O distance set to the smaller value did not lead to a change of lifetime within the error of the method[31].

The combined effect of all these factors has been observed experimentally, but to quantify the share of the individual mechanisms outlined above would require further studies.

This gradual increase raises the intriguing question whether the ICD efficiency reaches unity in bulk water. While the internal conversion process in water might still be very fast, the different solvation energies for singly and doubly charged species support the ICD decay, as outlined above. Our estimate of the potential energy curves in a liquid (Supplementary Fig. 5, Supplementary Note 4) indeed indicates that the ICD channel in a liquid never closes. Given the simple model we have used, we consider this result as preliminary, however. This question could be in principle resolved experimentally using the technique of coincident electron detection with a magnetic bottle spectrometer on a liquid microjet[40,41].

To summarize, in this work, we have unambiguously demonstrated that the efficiency of the Intermolecular Coulombic decay in water clusters subsequent to inner-valence ionization is significantly lower than unity. Based on calculations of the water dimer, we suggest that the dominant competing mechanism is a proton transfer between neighboring water units, taking place in less than 10 fs. Experimental results with isotopically substituted water clusters $(D_2O)_N$ further support the role of nuclear dynamics. The combination of coincidence experiments with high level ab initio dynamical simulations allow us to estimate the ICD lifetime in small water clusters to be in the range of 12 to 52 fs. This value is consistent with the ICD lifetime explicitly

calculated using the Fano-CI method. The ICD efficiency partially recovers when larger water clusters are investigated.

## Methods

**Experimental methods**. Experiments were carried out at the TGM4 and UE56/2-PGM1 beamlines of the synchrotron radiation source BESSY at Helmholtz-Zentrum Berlin, operating in single-bunch mode. Water clusters were produced by supersonic expansion of water (highly demineralized; conductivity $\approx 0.2$ $\mu$S cm$^{-1}$) and deuterated water vapor through a conical copper nozzle with an opening diameter of 80 $\mu$m, length 1.1 mm, and 15° opening angle[27]. Clusters formed in an expansion chamber, which was separated from the spectrometer chamber by a conical skimmer with an opening diameter of 1 mm. The background pressure in the spectrometer chamber during measurements was kept at $10^{-6}$ to $10^{-5}$ mbar. The formation of water clusters was verified with a quadrupole mass spectrometer (QMS) and recording of outer valence photoelectron spectra (PE) spectra. The mean size $\langle N \rangle$ of the cluster ensemble was determined from the expansion parameters using an empirical formula[42].

All experiments used a magnetic bottle spectrometer described in ref.[43] Briefly, it is designed with a drift tube of 0.6 m adapted to the BESSY bunch period of 800 ns. The magnetic guiding field has a strong inhomogeneous component produced by a permanent magnet below the interaction region (magnetic field strength $\approx 0.4$ T) and a weak homogeneous field in the drift region, induced by a solenoid wound around the drift tube. The spectrometer enables the detection of normal PE and electron–electron coincidence spectra with a high efficiency[44], determined as $\gamma$ ($E_{ICD}$) = 0.58(4) for the data presented. The coincident detection scheme allows separation of the initially ionized photoelectrons from the secondary electrons. The correlation between photoelectron and secondary electron, the latter being produced either by autoionization or by electron impact ionization in intracluster scattering processes, is readily observable in two-dimensional correlation maps (Supplementary Fig. 1).

**Computational methods**. The main model system for our calculations was the simplest water cluster, the water dimer, allowing us to accurately calculate the energetics of the $2a_1$ singly ionized and doubly ionized states.

The energies of the inner-valence-ionized state were calculated using second order perturbation theory based on the state averaged SA8-CASSCF reference wavefunction using 15 electrons in 8 orbitals (denoted as CASPT2(15,8)). In this restricted active space, the singly ionized state corresponding to $2a_1$ ionization of the donor molecule is the 7$^{th}$. We used the single state formulation of the CASPT2 scheme to avoid convergence issues. We have benchmarked the energetics of the inner-valence-ionized states with the Maximum Overlap Method (MOM)[45,46] based on different single reference methods such as MP2 and MP4 (Supplementary Note 4 for details). It is worth noting that both the MOM and restricted CAS approaches do not describe the satellite states, which we neglected in our treatment.

For single point calculations, we used the minimum geometry optimized at the MP2/6-31++G** level. Potential energy scans along the proton transfer coordinates were performed starting from this geometry by elongation of the respective O–H bond. To estimate the energetics in the liquid phase, we have calculated the $2a_1$ and doubly ionized states in the polarizable continuum models at the MP2/aug-cc-pVTZ level. The details of our protocol can be found in ref. [26], where it was validated for the core ionization of several small molecules. The $2a_1$ ionized state was obtained using the MOM method described above.

Since rigid potential energy surface scans can lead to misleading results, further dynamical calculations following the inner-valence ionization were performed on the CASPT2(15, 8)/6-31++G** potential energy surface described above. The dynamics was adiabatic, which is, however, appropriate given the restricted active space and the short-time dynamics following ionization from the $2a_1$ orbital of the donor molecule. The initial positions and velocities were sampled from the Wigner distribution of the ground vibrational state of the water dimer in a harmonic approximation at the MP2/6-31++g** level of theory. The normal modes below 500 cm$^{-1}$ were discarded, as the harmonic approximation is not valid in this region. The time step was 6 atomic units (0.15 fs) and the length of each simulation was 10 fs. The statistics was collected over 100 independent trajectories. All dynamical calculations were performed using the ABIN program[47].

To estimate the closing geometry of the ICD channel, the energy of both singlet and triplet doubly ionized states was calculated along each trajectory at the CASPT2(6,4)-SA6/6-31++G** level. When the energy of the inner-valence-ionized state drops below the lowest doubly ionized state, ICD can no longer take place.

The CASPT2 calculations were performed in the MOLPRO program[48,49] while the MOM calculations were done in the QCHEM program[50].

Few methods are available to calculate absolute electronic decay rates in clusters. For a direct computation of the total and partial decay rates of the water dimer, we employed the Fano-CI method. The details of this method are described elsewhere[31]. We utilized the cc-pVDZ basis set[51] on all atoms augmented with 5$s$, 5$p$, 5$d$ basis functions of the Kaufmann–Baumeister–Jungen (KBJ) type[52] centered on the oxygen atoms. The space of the decaying state included the $2a_1$ orbital of the donor or acceptor water molecule, whereas in the space of the final states we

included all outer valence orbitals $1b_2$, $3a_1$, and $1b_1$. The Fano-CI method lacks most of the dynamical correlation and therefore cannot reliably describe the delicate energetics of the initial and final ICD states in water dimer. To account for this deficiency, we have artificially restricted the number of final ICD states. In this article, we assume that only one final ICD singlet state and one final triplet state are available. Detailed benchmarking of this approach can be found in the Supplementary Note 2. The calculations were performed using our in-house Fano-CI code[53], while the necessary two-electron integrals were calculated using a modified version of the GAMESS-US code[54].

**Computer code availability**. Computer codes used for this work are part of common packages available freely or commercially; codes developed by the authors can be downloaded from the addresses given as refs.[47,53]

## Data availability

Raw data were generated at the large-scale facility for synchrotron radiation experiments of Helmholtz-Zentrum Berlin. Derived data supporting the findings of this study are available from the corresponding authors upon request.

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

## Acknowledgements

We thank Lasse Harbo and Evgeny Lugovoy for help with data acquisition. This project has received funding from the Research Executive Agency (REA) under the European Union's Horizon 2020 research and innovation program Grant Agreement No. 705515, and by the Deutsche Forschungsgemeinschaft via the Research Unit 1789. C.-M.S. acknowledges funding via the Swedish Foundation for International Cooperation in Research and Higher Education (STINT). P.S. and D.H. thank the Czech Science Foundation (Project Number 18-23756S). D.H. is a student of the International Max Planck Research School for 'Many-Particle Systems in Structured Environments'. N.S. acknowledges the financial support of the LabEx MiChem part of French state funds managed by the ANR within the 'Investissements d'Avenir' program under reference ANR-11-IDEX-0004-02. We thank HZB for the allocation of synchrotron radiation beamtime.

## Author contributions

M.F., P.S., and U.H. identified the problem; M.F. and U.H. conceived the experiment and performed the measurements of the isotope effect; C.R., C.-M.S., M.M., and U.H. performed measurements on water clusters; C.R., C.-M.S., and U.H. analyzed the data; D.H., T.M., N.S., and P.S. performed the calculations; C.R., D.H., C.-M.S., M.F., T.M., M.M., N. S., O.B., P.S., and U.H. discussed the data and contributed to writing of the manuscript.

## Additional information

**Competing interests:** The authors declare no competing interests.

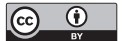

