## [Peer Review File · Nature Communications]

Reviewers' comments:

Reviewer #1 (Remarks to the Author):

The manuscript by Richter et al. describes a very nice work combining experimental and theoretical investigations on the intermolecular Coulombic decay (ICD) in water clusters. The experimental results seem reliable, and their interpretation is reasonable and well supported by high-quality quantum chemical calculations.

The authors explained that ICD has been observed and analyzed for rare gas clusters, and extended investigation to water clusters. Since ICD occurs in at least tens of femtoseconds, what would be potentially competing with it is nuclear dynamics of light atoms. Thus, it is expected that H atom elimination or proton transfer may compete with ICD, and there could be the deuterium isotope effect. I believe that they are predictable. The absolute quantum yield of ICD and its variation with the cluster size are shown in Figure 1. The authors explain that a proton transfer can occur and it competes with ICD using Fig. 2. On the other hand, while the readers can understand that proton transfer does occur after ionization to compete with ICD, in agreement with experimental results, they would not understand why the double-well potentials for the triplet and singlet states exhibit those shapes and how these potentials vary with the size of cluster. Moreover, while the key issue here is the origin of the cluster size dependence, discussion on page 11 is still highly speculative without strong support from their own computational study.

Although I think this is a very nice piece of work, some parts are predictable and the key question remains unanswered. Thus, I hesitate recommending it for publication on nature communications at this stage. On the other hand, this manuscript is already adequate for a specialized journal.

Reviewer #2 (Remarks to the Author):

—Overview—

In this work the authors argue that the contribution of ICD to the ionization of water clusters is less than previously thought, with proton transfer being the main competing process. They support this conclusion with a combination of experiment and theory. My impression of the manuscript is that is a comprehensive, transparent and thorough investigation.

The experimental side of the work involves measuring the kinetic energy statistics of electrons originating from irradiated water clusters, which are then identified as either coming from ICD processes or from others. From this the authors define an ICD efficiency, which would be unity if the only process present were ICD. The main difficulty in analysing the data seems to be the subtraction of a background coming from gaseous water molecules present in the jet. The authors apply several different background subtraction techniques, finding in all cases results displaying similar overall qualitative trends. The conclusion, which is well supported by the data presented, is that the ICD efficiency is less than 1. Based on some clues from these data and some hints from previous authors (refs 8 and 29), the authors identify proton transfer as the prime suspect for the decrease in the ICD efficiency.

It is at this point the paper switches from considering an experiment to looking at simulations. The authors calculate the potential energy curves for the water dimer, and show from there that the proton transfer decay channel is indeed open, and that the state after proton transfer is not ICD-open.

Using molecular dynamics techniques they determine the speed of the proton transfer process, which then limits the time available for ICD to happen, and is dependent on the specific isotope at hand (unlike ICD itself which is purely electronic).

The authors then address the increase of the ICD efficiency when larger clusters are used. They present three different explanations; increased number of neighbours, cluster-size induced level shifts, and varying bond distance. They do not make quantitative estimates for any of these effects, rather they are presented as suggestions which may work independently or together in order to explain the observed trend.

—Comments—

The only major issue I have with the article is with its structure. While it is mostly well-structured, the section beginning halfway down page 11 (“We now would like to address the increase in ICD efficiency...”) reads like a last-minute addition (perhaps suggested by a previous referee), and seems disconnected from the rest of the manuscript. This is because the title and main thrust of the manuscript both rightly focus on one very specific fact (proton transfer quenches ICD in water), while the section on page 11 has no relation to this, as far as I can tell. The start of the section beginning halfway down page 11 should be specifically referred to as an aside, moved to the supplementary material, or removed entirely. The same goes for the main panel of Fig. 1c (not the inset), which essentially says that the ICD efficiency goes up with cluster size. As the authors later state, both the proton transfer rate and the ICD rate are expected to increase with increasing cluster size, so the main panel of Fig 1c has no bearing on the main statement given by the manuscript’s title. I have no problem with including this cluster size-dependence as an aside. However, my feeling is that main panel of Fig 1c should either be moved into the section on page 11 referred to above, or into the supplementary or deleted altogether. This would also allow the inset of Fig 1c to be emphasised more, which would be strengthen the manuscript since it has a more direct relevance to the main arguments presented.

Provided the authors can address these queries to my satisfaction (alongside the minor issues listed below), I would be happy to recommend publication in Nature Communications.

—Minor issues—

I have some other minor queries which I would like addressed. In no particular order:

1) The degree of condensation c is a critical factor in comparing figures 1a and 1b (which are those that most strongly support the main conclusion of the paper). In my first reading of the manuscript it was unclear whether this quantity was measured or fitted, and if it were the latter the conclusions of the paper would be seriously undermined. In reality it is the former — the authors have robustly measured c for their particular setup. This strength of the analysis should be highlighted more prominently in the discussion of figure 1, either in the caption or the text or both.

2) ICD to my mind means interatomic (not necessarily intermolecular) Coulombic decay. While this may be a matter of taste, the authors should acknowledge that there is more than one possible meaning of the acronym ICD.

3) I find the use of the word “entanglement” on page 3 strange. To my mind this word should be reserved for entanglement of the quantum kind, while here it seems to me as if the authors mean it in the more everyday sense of combining two objects (or concepts) together. If the authors really do mean quantum entanglement then this should be explained, and if not they should change the word

“entanglement” to a suitable alternative (“intertwining”, for example).

Reviewer #3 (Remarks to the Author):

The present manuscript reports on the Intermolecular Coulombic decay (ICD) in water clusters with a mean size between 5 to about 250 molecules. This process is of primary importance as it may have an important role in radiation treatment as a source of slow electrons which are known to be highly damaging. The authors report a decrease of the ICD efficiency, compared to that of rare gas clusters which amounts to 100%, and try to explain it. The answer comes from a competing process namely the proton transfer which may avoid the ICD process to occur. This statement is convincingly argued not only by energy level calculations but also by the evolution of the ICD efficiency when deuterated water is used or when the cluster size is increased. This is a very important achievement as it interests a wide range of scientific topics from molecular science to radiobiology.

The paper is clearly written and argued.

So in conclusion, I have no corrections to suggest and I would recommend this paper for publication in Nature Communications.

Reviewer(s)' Comments to Authors

Reviewer 1 Comments:

The manuscript by Richter et al. describes a very nice work combining experimental and theoretical investigations on the intermolecular Coulombic decay (ICD) in water clusters. The experimental results seem reliable, and their interpretation is reasonable and well supported by high-quality quantum chemical calculations.

We thank the reviewer for the positive evaluation of our work.

The authors explained that ICD has been observed and analysed for rare gas clusters, and extended investigation to water clusters. Since ICD occurs in at least tens of femtoseconds, what would be potentially competing with it is nuclear dynamics of light atoms. Thus, it is expected that H atom elimination or proton transfer may compete with ICD, and there could be the deuterium isotope effect. I believe that they are predictable. The absolute quantum yield of ICD and its variation with the cluster size are shown in Figure 1. The authors explain that a proton transfer can occur and it competes with ICD using Fig. 2. On the other hand, while the readers can understand that proton transfer does occur after ionization to compete with ICD, in agreement with experimental results, they would not understand why the double-well potentials for the triplet and singlet states exhibit those shapes and how these potentials vary with the size of cluster. Moreover, while the key issue here is the origin of the cluster size dependence, discussion on page 11 is still highly speculative without strong support from their own computational study.

The main point of the reviewer is that the discussion of the size dependence – which itself is of much interest – should be improved in the revised manuscript. We believe that some of the reviewer's discomfort has been caused by a less-than-optimal structure of our original manuscript, which led to an unclear separation between two facets of our work: 1. The less than unity efficiency of ICD and its cause by competition with proton transfer, and 2. The variation of the less-than-unity ICD efficiency with cluster size. These are both important and of course interrelated, but also clearly distinguishable. We agree with the reviewer that our explanation of the size-dependence was more speculative than our explanation of the proton transfer as such.

In order to improve our manuscript we have taken two measures. 1. The discussion of the size dependence has been moved to the end of the discussion section, so that it does no longer interfere with the preceding discussion of the proton transfer and can be appreciated as what it

is: An experimental result, which we present and for which we suggest plausible explanations. At present, we cannot offer a fully unambiguous explanation for the origin of the dependence. We consider three possible effects – and to some extent we indeed can only speculate on their relative role. This is, however, a fully scientific approach – we do not come to definite conclusions unless we have a solid evidence for them.

2. Our tentative conclusions were already supported by some calculations of the energetics of the differently charged species. This part has now been extended and we have added the system's potential curves calculated within a dielectric continuum model (as an affordable simulation of a large cluster or infinite liquid) to the Supplementary Material. These calculations corroborate a trend of increasing ICD energy also observed experimentally, leading also to an increase in the time over which the ICD channel stays open. This lends credibility to one of the mechanisms we have lined out as an explanation for the observed size dependence. Nevertheless, the other mechanisms can also be operative even if we have a reason to believe their role is less important.

We now comment on some other points raised critically by the reviewer:

We cannot fully agree with the statement that the limited efficiency of the ICD due to proton transfer as the main competitive mechanism is a predictable fact. In fact, we believe (together with reviewer 2) that this is the main message of our work, fully supported by experiment and theory. Note that the ICD efficiency for isoelectronic neon clusters is 100% and it is by no means obvious that the efficiency should be rather low for water clusters. Based on analogy (no actual data available so far!), we could anticipate that the ICD lifetime is relatively long to allow for a motion of the light atoms. In this sense, it is indeed predictable that proton motion takes place within the ICD lifetime. However, it was not clear at all that it would close the ICD channel. This is only dictated by the particular shapes of the potential energy surfaces which are not easily predictable. The present study is the first one to reveal the details of ICD in molecular systems. We can thus also provide an independent estimate of the ICD lifetime based on the isotope effect (another 'first').

The reviewer mentions that for readers it is hard to understand why the potential curves exhibit the double-well shapes we have found. This indeed was not sufficiently explained, and we added the sentence:

“Both singlet and triplet doubly ionized ground states correspond to the final states of the ICD process, which are charge separated or '1h1h' states. The corresponding potential energy curves exhibit a double-well shape: The left minimum correspond to the $\text{H}_2\text{O}^+ \dots \text{H}_2\text{O}^+$ structure while the right one represents the $\text{OH}^+ \dots \text{H}_3\text{O}^+$ structure. The maximum on the curves correspond to the transient $\text{HO} \dots \text{H}^+ \dots \text{H}_2\text{O}^+$ species.”

Although I think this is a very nice piece of work, some parts are predictable and the key question remains unanswered. Thus, I hesitate recommending it for publication on nature communications at this stage. On the other hand, this manuscript is already adequate for a specialized journal.

Summarizing, we respect the foresight of the reviewer but do not concur that the competition of ICD with proton transfer is *a priori* so clear that its description is of interest only to a very narrow community. The reviewer has pointed to certain weaknesses in the presentation of our data. We have addressed all the points raised and we added new calculations to strengthen our conclusions. We believe that the work will now seem fully convincing for the reviewer.

Reviewer 2 Comments:

In this work the authors argue that the contribution of ICD to the ionization of water clusters is less than previously thought, with proton transfer being the main competing process. They support this conclusion with a combination of experiment and theory. My impression of the manuscript is that is a comprehensive, transparent and thorough investigation.

We thank the reviewer for the positive evaluation of our work.

The experimental side of the work involves measuring the kinetic energy statistics of electrons originating from irradiated water clusters, which are then identified as either coming from ICD processes or from others. From this the authors define an ICD efficiency, which would be unity if the only process present were ICD. The main difficulty in analysing the data seems to be the subtraction of a background coming from gaseous water molecules present in the jet. The authors apply several different background subtraction techniques, finding in all cases results displaying similar overall qualitative trends. The conclusion, which is well supported by the data presented, is that the ICD efficiency is less than 1. Based on some clues from these data and some hints from previous authors (refs 8 and 29), the authors identify proton transfer as the prime suspect for the decrease in the ICD efficiency.

It is at this point the paper switches from considering an experiment to looking at simulations. The authors calculate the potential energy curves for the water dimer, and show from there that the proton transfer decay channel is indeed open, and that the state after proton transfer is not

ICD-open. Using molecular dynamics techniques they determine the speed of the proton transfer process, which then limits the time available for ICD to happen, and is dependent on the specific isotope at hand (unlike ICD itself which is purely electronic).

The authors then address the increase of the ICD efficiency when larger clusters are used. They present three different explanations; increased number of neighbours, cluster-size induced level shifts, and varying bond distance. They do not make quantitative estimates for any of these effects, rather they are presented as suggestions which may work independently or together in order to explain the observed trend.

Comments

The only major issue I have with the article is with its structure. While it is mostly well-structured, the section beginning halfway down page 11 (“We now would like to address the increase in ICD efficiency...”) reads like a last-minute addition (perhaps suggested by a previous referee), and seems disconnected from the rest of the manuscript. This is because the title and main thrust of the manuscript both rightly focus on one very specific fact (proton transfer quenches ICD in water), while the section on page 11 has no relation to this, as far as I can tell. The start of the section beginning halfway down page 11 should be specifically referred to as an aside, moved to the supplementary material, or removed entirely. The same goes for the main panel of Fig. 1c (not the inset), which essentially says that the ICD efficiency goes up with cluster size. As the authors later state, both the proton transfer rate and the ICD rate are expected to increase with increasing cluster size, so the main panel of Fig 1c has no bearing on the main statement given by the manuscript’s title. I have no problem with including this cluster size-dependence as an aside. However, my feeling is that main panel of Fig 1c should either be moved into the section on page 11 referred to above, or into the supplementary or deleted altogether. This would also allow the inset of Fig 1c to be emphasised more, which would be strengthen the manuscript since it has a more direct relevance to the main arguments presented.

We agree with the reviewer that the most essential result reported in the manuscript is the fact that the ICD efficiency is limited, and it is the proton transfer which closes this channel. The fact that we have carried out a size-dependent study of the ICD efficiency however also is a novelty, and reviewer 1 assessed this as the ‘key issue’. In our opinion this underpins that these two aspects of our story both are important. Admittedly however, we have a solid understanding *why* the ICD efficiency is as low as we find it experimentally, while for the size dependence we have identified one of the acting mechanism, without at present being able to rule out two other factors.

In order to address the shortcomings that reviewers 1 and 2 had concerning the structure of our manuscript, we have taken the following measures:

1. The first figure in the manuscript now focusses fully on the proton transfer issue, by showing typical experimental data in two panels, and the isotope effect on the efficiency as a third panel. The size dependence (formerly panel c in Fig. 1) is now presented as a separate figure, Fig. 2.
2. We present both figures at the beginning of the results section, as we use data for the smallest clusters we have measured for comparison to our calculations. However, we defer further explanation of Fig. 2 to the end of the 'Discussion' section, which, we believe, leads to a more natural flow of the narrative. This is also clearly explained to the readers.
3. Some redundancy in the discussion of former Fig. 1 (now Figs. 1 and 2) has been removed.

Provided the authors can address these queries to my satisfaction (alongside the minor issues listed below), I would be happy to recommend publication in Nature Communications.

Minor issues

I have some other minor queries which I would like addressed. In no particular order:

1) The degree of condensation c is a critical factor in comparing figures 1a and 1b (which are those that most strongly support the main conclusion of the paper). In my first reading of the manuscript it was unclear whether this quantity was measured or fitted, and if it were the latter the conclusions of the paper would be seriously undermined. In reality it is the former --; the authors have robustly measured c for their particular setup. This strength of the analysis should be highlighted more prominently in the discussion of figure 1, either in the caption or the text or both.

We thank the reviewer for pointing this out. In the discussion of Fig. 1 we have added a sentence reading:

“ c was determined at each data point from outer valence electron spectra, with an example shown in Supplementary Fig. 2, and corresponding results in Supplementary Fig. 3.”

2) ICD to my mind means interatomic (not necessarily intermolecular) Coulombic decay. While this may be a matter of taste, the authors should acknowledge that there is more than one possible meaning of the acronym ICD.

We agree with this remark, although the publication that coined the notion of ICD (Ref. (1)) was titled ‘intermolecular decay of clusters ...’ We surmise it became known as Interatomic Decay because of the bulk of experiments on rare gas clusters. We made a note on this issue on page 2, and corrected the following sentence. The passage now reads:

“Depending on the constituents of the system under consideration, ICD was taken to mean either Intermolecular or Interatomic Coulombic Decay.

Interatomic Coulombic Decay was first observed in rare gas clusters following inner valence electron ionization”

Along the same lines, the last sentence of the 2nd par. on p. 3 has been changed to:

“Depending on the particular situation, the nuclear dynamics may suppress or accelerate the interatomic or intermolecular autoionization processes.”

3) I find the use of the word “entanglement” on page 3 strange. To my mind this word should be reserved for entanglement of the quantum kind, while here it seems to me as if the authors mean it in the more everyday sense of combining two objects (or concepts) together. If the authors really do mean quantum entanglement then this should be explained, and if not they should change the word “entanglement” to a suitable alternative (‘intertwining’, for example).

Indeed, we should avoid using terms which have their specific meaning in different fields. We replaced the expression by simple “coupling”.

Reviewer 3 Comments:

The present manuscript reports on the Intermolecular Coulombic decay (ICD) in water clusters with a mean size between 5 to about 250 molecules. This process is of primary importance as it may have an important role in radiation treatment as a source of slow electrons which are known to be highly damaging. The authors report a decrease of the ICD efficiency, compared to that of rare gas clusters which amounts to 100%, and try to explain it. The answer comes from a competing process namely the proton transfer which may avoid the ICD process to occur. This statement is convincingly argued not only by energy level calculations but also by the evolution of the ICD efficiency when deuterated water is used or when the cluster size is increased. This is a very important achievement as it interests a wide range of scientific topics from molecular science to radiobiology.

The paper is clearly written and argued.

So in conclusion, I have no corrections to suggest and I would recommend this paper for publication in Nature Communications.

We thank the reviewer for the positive evaluation of our work and for supporting the publication of our work in Nature Communications.

Besides the changes outlined above, at some places we have done minor edits to make the text clearer or easier to read, and to adapt the style to the requirements of Nature Communications. A paper that came to our attention while carrying out the edits (Ref. (37)) is briefly commented on. Finally, we have changed the title of the manuscript to 'Proton transfer and Intermolecular Coulombic Decay compete in water' to more accurately describe the findings of our work.

REVIEWERS' COMMENTS:

Reviewer #1 (Remarks to the Author):

The authors revised the manuscript properly. It is now acceptable.

Reviewer #2 (Remarks to the Author):

My major concern in my previous review was the structure, which has been substantially improved by splitting the results and discussion into two separate sections. This aids to readability since it clearly separates the unambiguously reasoned conclusions from the more speculative aspects (not a criticism) of the work. My minor concerns have been all also addressed. Therefore, I recommend publication in Nature Communications.